# Cytotoxicity of Metal Ions Released from NiTi and Stainless Steel Orthodontic Appliances, Part 1: Surface Morphology and Ion Release Variations

**DOI:** 10.3390/ma16114156

**Published:** 2023-06-02

**Authors:** Mirna Petković Didović, Ivana Jelovica Badovinac, Željka Fiket, Jure Žigon, Marijana Rinčić Mlinarić, Gordana Čanadi Jurešić

**Affiliations:** 1Department of Medical Chemistry, Biochemistry and Clinical Chemistry, Faculty of Medicine, University of Rijeka, B. Branchetta 20, 51000 Rijeka, Croatia; 2Faculty of Physics and Centre for Micro- and Nanosciences and Technologies, University of Rijeka, Radmile Matejčić 2, 51000 Rijeka, Croatia; 3Division for Marine and Environmental Research, Ruđer Bošković Institute, Bijenička 54, 10000 Zagreb, Croatia; 4Biotechnical Faculty, University of Ljubljana, Jamnikarjeva 101, 1000 Ljubljana, Slovenia; 5Private Orthodontic Practice, Katarine Zrinske 1b, 23000 Zadar, Croatia

**Keywords:** orthodontic appliances, ion elution, surface properties, NiTi, stainless steel, ligatures, brackets, bands

## Abstract

Despite numerous studies on ion release from orthodontic appliances, no clear conclusions can be drawn due to complex interrelations of multiple factors. Therefore, as the first part of a comprehensive investigation of cytotoxicity of eluted ions, the objective of this study was to analyze four parts of a fixed orthodontic appliance. Specifically, NiTi archwires and stainless steel (SS) brackets, bands, and ligatures were immersed in artificial saliva and studied for morphological and chemical changes after 3-, 7-, and 14-day immersion, using the SEM/EDX technique. Ion release profiles were analyzed for all eluted ions using inductively coupled plasma mass spectrometry (ICP-MS). The results demonstrated dissimilar surface morphologies among parts of the fixed appliance, due to variations in manufacturing processes. The onset of pitting corrosion was observed for the SS brackets and bands in the as-received state. Protective oxide layers were not observed on any of the parts, but adherent layers developed on SS brackets and ligatures during immersion. Salt precipitation, mainly KCl, was also observed. ICP-MS proved to be more sensitive than SEM/EDX and exhibited results undetected by SEM/EDX. Ion release was an order-of-magnitude higher for SS bands compared to other parts, which was attributed to manufacturing procedure (welding). Ion release did not correlate with surface roughness.

## 1. Introduction

Although there are numerous studies on toxicity and ion release from orthodontic materials, the results are not consistent and no clear conclusions can be drawn. The reason for this predicament lies in the complex interconnections of numerous factors: besides the obvious ones such as chemical composition of the alloy and the type of medium used for elution, there are others such as phase structure of an alloy, thermal history, surface roughness, the presence of adherent layers, morphological features (pits, crevices), manufacturing procedure, welding/soldering, and contact with other alloys [1,2,3,4]. The situation is further exacerbated by the possibility that surface composition may differ from the bulk or the nominal composition, that small differences in surface preparation may induce large differences in corrosion stability, or that parts made from the same alloy may differ from vendor to vendor [5,6,7,8]. To quote Wataha and Schmalz: Research in this area has generated as many questions as it has answered [1].

The oral cavity, with its own physiological, thermal, microbiological, and biochemical characteristics, forms an ideal environment for the biodegradation of dental materials. Biodegradation itself is strongly favored by foods and beverages that further promote corrosion. A diet rich in sodium chloride and acidic carbonated beverages provides a regular supply of corrosive substances. Modern orthodontic therapy is performed with a fixed appliance, the components of which are made of various alloys. Although orthodontic appliances are only placed (not implanted) in the oral cavity and do not seem to have such a close relationship with the body tissues, oral corrosion leads to the release of many different ions. These ions enter the biological system of the oral cavity and affect the surrounding tissues [9]. Numerous side effects can occur, ranging from hypersensitivity reactions and tissue proliferation to cytotoxicity and genotoxicity [3,4,10]. Local cytotoxicity as the first effect has been demonstrated in epithelial cells and fibroblasts of the periodontal ligament [11,12]. Hypersensitivity reaction occurs as a second effect when metal ions interact with human proteins such as haptens [13]. The most common form of allergic reaction to dental biomaterials in the oral cavity is contact allergic reaction, late-type hypersensitivity reaction, or cell-mediated hypersensitivity (type IV) [14].

Under normal circumstances, the pH in the oral cavity is not stable and changes ac-cording to the intake of food and drink, but homeostasis returns to neutral values. Maintaining oral hygiene is extremely difficult during fixed orthodontic therapy. The accumulation of plaque and inflammatory processes in the gingiva lead to a drop in pH and the formation of an acidic environment that promotes the degradation of dental materials [15]. For this reason, it is interesting to study the oral conditions in patients with poor hygiene.

This study is the first part of a wider investigation of the toxic effect of metal ions released from orthodontic appliances on gingival fibroblasts, gastrointestinal tract cell lines, and yeasts. For all the reasons mentioned above (biocompatibility, structure of the materials used, processes in the oral cavity, sensitivity of patients, and many others), we wanted to investigate the actual appliance (with the most commonly used parts) and use it for the in-depth toxicity study.

The main orthodontic base metal alloys commonly used for wires, brackets, and other orthodontic appliances are stainless steel (SS), cobalt−chromium, nickel−titanium (NiTi), and beta titanium [4,10,16]. In this study, NiTi archwires and three types of SS parts (brackets, bands, and ligatures) were used because together they form a fixed orthodontic appliance. SS alloys are ferrous alloys with a Cr content of about 12% or more. In addition to Cr, they also contain Ni and, in small amounts, Mo, W, N, Cu, C, Ti, S, Si and Mn. There are numerous grades of stainless steel with varying contents of the major metals (chromium, nickel, and molybdenum). Cr forms a thin oxide that adheres to the surface (called passivation) and increases corrosion resistance by blocking attack on the underlying metal [9,17]. A higher Ni content of about 11% or more is also associated with good corrosion resistance, by competing with the chromium to form salts, making more chromium available for passivation [9]; Ni also improves processing capabilities (welding, cold heading, cold forming, polishing, etc.) [10,18]. Based on microscopic structures, SS can be divided into austenitic martensitic, ferritic, and duplex (ferrite + austenite) steels. Austenitic steels have better corrosion resistance than ferritic and martensitic steels.

Although three out of four parts used in this study are made of SS, they are manufactured differently to serve their specific purpose in the orthodontic appliance. The brackets are manufactured in one piece using the metal injection molding (MIM) technique. This technique ensures that galvanic corrosion does not occur, but also results in higher porosity [10]. To achieve a perfect fit to the tooth, the bands are highly polished on the outside and roughened on the inside in a special process [17]. The surface finish has a relatively large influence on the corrosion resistance of the steel, so that the polished surface usually has a higher corrosion resistance than the rough surface. During orthodontic tooth movement, the archwires are attached to the bracket slots with SS ligatures. An automatic bending machine and a pure stainless steel without coating or surface modification are used to manufacture these ligatures [10].

Unlike stainless steel, orthodontic NiTi alloy is a shape memory alloy with unique properties: superelasticity and shape memory effect. The memory properties of NiTi-based alloys are limited to a narrow compositional range near their equimolar (equiatomic) composition. Alloy compositions that deviate from the equimolar ratio lead to the precipitation of second-phase particles (such as Ti_2_Ni, Ti_3_Ni_4_, Ni_3_Ti), which significantly alter the fatigue and functional properties of the alloy [19]. Although the nickel-rich composition was chosen because it gives superelastic properties to the wires, it is also associated with problems when used in orthodontics. The high nickel content (47–50%) is related to nickel leaching, while the surface roughness is related to corrosion behavior, plaque accumulation, and the effectiveness of the sliding mechanism.

This paper is organized as follows: First, the SEM micrographs of the individual parts of the fixed appliance are qualitatively analyzed in the as-received state and after 3-, 7-, and 14-day immersion in artificial saliva. Then, the chemical compositions of specific surface sites and morphological features are analyzed semi-quantitatively using energy dispersive X-ray (EDX) spectroscopy. Next, the eluted concentrations of all released ions are presented and analyzed. The objective of the study was to (a) investigate the differences in surface morphology and chemical composition between different parts of the fixed orthodontic appliance; (b) analyze the morphological and chemical changes induced by the immersion in artificial saliva; (c) obtain the ion release profiles of each part; and (d) establish potential correlations between the changes in surface morphology and ion release profiles.

## 2. Materials and Methods

### 2.1. Orthodontic Appliances

“Rematitan” archwires (rematitan^®^ LITE ideal arches, *φ* 0.43 × 0.64 mm/17 × 25, Dentaurum) made of an alloy of nickel and titanium (40–50%) were used in the study. The bands used (dentaform, tooth 36, size 23/Roth 22, Dentaurum) were made of stainless steel. Brackets (equilibrium^®^ 2, *φ* 0.56 × 0.76 mm/22 × 30, Roth 22, Dentaurum) and ligatures (remanium^®^, short, soft, *φ* 0.25 mm/10, Dentaurum) were also made of stainless steel. The detailed composition of all devices is shown in Table 1.

Alloy composition can be expressed in 2 ways: weight percent (wt%) and atomic percent (at%). Although the weight percentage is widely used and standardized, the atomic percentage is more suitable for predicting the effects of released ions on biological systems [3,20].

All SS parts are made of austenitic steel, which is known for its good corrosion resistance in many different environmental conditions. Although they are SS, they differ in a grade that meets specific requirements that each device should have (intergranular corrosion resistance, cold formability, deep drawability and weldability, easy and precise placement, elasticity, etc.). Stainless steel alloys 1.4303 (AISI 305; X4CrNi18-12) and 1.4551 (X5CrNiNb19-9) were used to manufacture the bands in the form of prefabricated sheet metal strips, smooth on one side and rough on the other. An orthodontic band also has a tunnel-shaped sleeve through which the archwire is pulled and which is attached to the band by spot welding. The ligatures were made of stainless steel 1.4310 (AISI 301; X10CrNi18-8) with an automatic bending machine, without coating or surface treatment. The metastable austenitic steel grade 1.4301 (AISI 304; X5CrNi18-10) is used to manufacture the brackets, and the bracket is produced as a one-piece bracket by metal injection molding (MIM) [10].

### 2.2. Experimental Media

For the experiment, artificial saliva with a pH of 4.8 was used, prepared by the Tani-Zucchi method. In this recipe, 1.5 g/L KCl, 1.5 g/L NaHCO_3_, 0.5 g/L NaH_2_PO_4_ × H_2_O, 0.5 g/L KSCN, and 0.9 g/L lactic acid were mixed [21]. The pH value chosen for the experiment reflects the pH value found in patients with pure oral hygiene [15].

### 2.3. Methods

#### 2.3.1. Preparation of the Artificial Saliva Test Samples/Eluates

We prepared the samples (eluates in artificial saliva) according to the current ISO standard (ISO 10993-5:2009). This standard describes test methods for evaluating the in vitro cytotoxicity of medical devices. Briefly, one orthodontic archwire, ten brackets, ten ligatures, and two bands (each type of appliance separately) were immersed in 20.0 mL of artificial saliva and autoclaved at 121 °C for 15 min (CertoClav, Leonding, Austria) and then incubated under sterile conditions on a rotary shaker (37 °C, 100/min, Unimax 1010, Heidolph, Germany) for 3, 7, and 14 days, respectively. Five samples were prepared for each time interval and all appliance combinations. After the experimental period, the samples of the same time interval were combined in a new sterile tube. From each sample, 7.0 mL was taken to determine the released metal ions using an inductively coupled plasma mass spectrometer. The remainder of each sample was used for further experiments with yeast/cells. The surface of the appliances used was examined microscopically.

#### 2.3.2. Surface Morphology Determination

The surface roughness of used devices was examined on two examples of each device on two areas with a LEXT OLS5000 confocal laser scanning microscope (abbreviated as CLSM) (Olympus, Tokyo, Japan). The analyses at 50× magnification provided the scanned area of 257 × 257 μm^2^. The areas were scanned with a 405 nm laser light with a maximum lateral resolution of 0.12 μm. The OLS50-S-AA software (Olympus) was used to reconstruct 3-dimensional images of the analyzed surfaces and to calculate the arithmetical arithmetic mean surface roughness, S_a_, and the developed interfacial area ratio, S_dr_.

The appearance of the metal surface of used appliances was also studied with a scanning electron microscope (SEM, Jeol JSM-7800F, Tokio, Japan) by collecting secondary electrons with an electron beam accelerating voltage of 10 kV and a working distance of 10 mm. The energy dispersive X-ray EDX spectrometer (X-Max 80, Oxford Instruments, Abingdon, UK) in SEM was used to study the elemental composition of representative samples with a beam acceleration voltage of 12 kV. At least two specimens of each tested elution time device were tested at three surface points, with at least three additional EDX spectroscopic examinations.

Each element is standardized by means of a standard label (for aluminum Al_2_O_3_ is used, for argon Ar(V), for calcium wollastonite, for carbon vitamin C, for chlorine NaCl, for chromium Cr, for copper Cu, for iron Fe, for magnesium MgO, for manganese Mn, for nickel Ni, for phosphorus GaP, for oxygen and for silicon SiO_2_, for potassium KBr, for radon, Rh, and for sodium, albite).

#### 2.3.3. Multi-Element Analysis of Artificial Saliva Samples before and after Immersion (ICP-MS)

A 2-fold dilution of prepared samples was performed, then samples were acidified with 2% (*v*/*v*) HNO_3_ (65%, Fluka, Steinheim, Switzerland), and In (1 g/L) was added as an internal standard. High-resolution inductively coupled plasma mass spectrometry (HR-ICP-MS; Thermo Scientific, Bremen, Germany) was used to analyze the prepared samples for multiple elements. Fiket et al. [22] describe the method in detail. Quantification was performed using an external calibration. Metal detection limits were 0.1 μg/L.

The multi-element standard (Analytika, Prague, Czech Republic) containing Al, As, Ba, Be, Bi, Cd, Co, Cr, Cs, Cu, Fe, Li, Mn, Mo, Ni, Pb, Se, Sr, Ti, Tl, V, and Zn was diluted accordingly. We added single-element standard solutions of U (Aldrich, Milwaukee, WI, USA), Rb (Aldrich, Milwaukee, WI, USA), Sn (Antika, Prague, Czech Republic), and Sb (Analytika, Prague, Czech Republic). For Ca determination, an elemental reference standard was used (Fluka, Neu-Ulm, Germany). All samples were analyzed for total concentration of n elements (Al, As, Ba, Be, Bi, Ca, Cd, Co, Cr, Cs, Cu, Fe, Li, Mn, Mo, Ni, Pb, Rb, Sb, Se, Sn, Sr, Ti, Tl, U, V, Zn). Analytical quality control was performed by analyzing both the blank sample and certified reference material (SLRS-4, NRC, Canada). All elements showed good agreement between their analyzed and certified concentrations within their analytical uncertainties (±10%).

#### 2.3.4. Statistical Analyses

For comparison of surface roughness parameters S_a_ and S_dr_, the Mann–Whitney U test was used, whereas for comparison of eluted concentration of major metal ions from parts of the orthodontic appliance, analysis of variance (ANOVA) was used together with the post hoc Tukey HSD test (using Statistica data analysis software system, version 13.4.04; Tibco Software Inc, Palo Alto, CA, USA).

## 3. Results and Discussion

### 3.1. Surface Morphology—Qualitative SEM Analysis

The differences in the surface morphology were first examined by qualitative SEM analysis. Representative images of the samples in an as-received state and after immersion at 37 °C in artificial saliva for 3, 7, and 14 days are shown in Figure 1, Figure 2, Figure 3 and Figure 4, respectively.

#### 3.1.1. Pristine Samples

SEM micrographs showed evident differences in the surface texture between samples, which can be attributed to their different chemical compositions and specific manufacturing processes. The surface of NiTi archwires and ligatures (Figure 1a,d) contained striations and crevices (machining marks [23]), with notable amounts of debris accumulated within them in the case of NiTi wires. The areas between striations appeared less smooth for ligatures compared to NiTi wires. The micrographs also implied that there was no protective oxide layer present on the NiTi surface, since Ti- and Ni-oxides would exhibit highly recognizable surface textures [24]. This further means that for the NiTi wires studied here the oxide layer was removed by mechanical polishing [5].

The surface of the brackets and the bands (Figure 1b,c) contained other types of heterogeneities, most notably the black spots/pits ranging from submicron to several microns in size. The spots resembled corrosion cavities, thus indicating that the corrosion process has started before the immersion in artificial saliva. The onset of pitting corrosion on new orthodontic appliances has been reported in other studies as well [9,25]. On the other hand, no corrosion pits were noticed on the NiTi wires and ligatures, even though the ligatures were made of a very similar type of SS as the brackets (Table 1). Data obtained from the manufacturer confirmed that no coatings or additives were used on the ligatures (private correspondence); hence, the differences probably arose from variations in the manufacturing process. Indeed, it has been demonstrated that the polishing of NiTi wires to a uniform finish reduces the corrosion rate [9,26].

Alongside the corrosion pits, the brackets contained very few other morphological features, except numerous flower-like formations on the underside (depicted in the addendum in Figure 1b), arranged in a formation that reflected one of the possible shapes of the base morphology [27], with a purpose of improving the adhesion to the tooth [28]. The bands (Figure 1c) contained large, irregularly shaped cavities several tens of micrometers in size, with smooth areas in between. The lower side (the addendum in Figure 1c) contained a markedly smaller proportion of smooth areas. The differences between the two sides, also visible macroscopically, are the result of intentionally lower polishing level of the lower side, again to improve the adhesion to the tooth.

The SEM micrographs also showed that all parts except the ligatures contained small amounts of approximately micrometer-sized white crystal deposits. Similar impurities were found in other studies, located in the corrosion pits [25] or elsewhere on the surface [29].

#### 3.1.2. Immersion after 3 Days

After 3-day immersion, the surface of the NiTi archwire remained smooth and uniform, and exhibited minor surface contamination, mainly in the grooves (Figure 2a,b). These entities varied in size and morphology, indicating differences in chemical composition, but the size of individual crystals did not exceed ~1 μm. Crevices and striations were similar to those in the pristine state. In general, almost the entire archwire surface remained unaffected.

Three days of immersion did not notably enhance the pitting corrosion in SS brackets, nor did it result in deposits or adherent material. However, the protrusion of crystal clusters from the bulk was observed. Higher magnification images (representative image given as upper addendum in Figure 2d) further indicated that the small white entities were not the adhering material. Besides those, the lower bracket surface, containing areas of various morphologies (addendums in Figure 2c,d), also showed no notable changes.

On the contrary, the SEM micrographs of SS bands (Figure 2e,f) showed prominent changes and a very small portion of the unaffected surface. Most of the surface was covered with an adherent layer that was not detected on other parts of the fixed appliance. This layer can be recognized as an oxide protective layer, which is common on SS products [30,31,32,33]. Lighter colored deposits were observed on the top of the protective layer and in the areas in between, which were reminiscent of layer debris [34].

Black pits were visible on the surface of SS ligatures (Figure 2g,h), indicating the onset of corrosion. An increased number of white crystal precipitates was also observed.

#### 3.1.3. Immersion after 7 Days

After 7-day immersion, no signs of localized corrosion or oxide layer formation were observed on the NiTi surface (Figure 3a,b). The quantity of deposits increased, both in number and size.

SEM micrographs (Figure 3c,d) indicated changes on the surface of SS brackets. Two main features were observed on the upper surface: the large amount of irregularly shaped large cavities (>10 μm) and large, regularly shaped crystal deposits, up to 10 μm in size. The lower surface appeared to be less affected, as indicated by the addendums on Figure 3c,d. The amount and the size of the corrosion pits appeared similar to those observed after 3-day immersion.

For the SS bands (Figure 3e,f), the surface appeared more homogeneous after 7 days in AS in comparison to the 3-day surface (in terms of oxide layer patches and debris), indicating the development of the protective layer. The layer contained small cavities of submicron size.

The ligatures (Figure 3e,f) exhibited higher amounts of deposits mainly localized in the crevices. The crystals had a characteristic flower-like morphology, as shown in the addendum in Figure 3h.

#### 3.1.4. Immersion after 14 Days

After 14-day immersion, development was observed on the surface of the NiTi archwires (Figure 4a,b). New crevices and grooves were formed, and the surface—both on the inside and the outside of the surface irregularities—was abundant with white precipitates/deposits of various sizes. No signs of pitting corrosion were observed.

Pitting corrosion appeared to be slightly advanced on the SS bracket surface (Figure 4c,d), but this should be confirmed by quantitative methods. For comparison, Frois et al. [31] found no relevant morphological changes on the surface even after 30 days of immersion. New deposits of specific morphology formed, having a size of a few tens of micrometers. Larger white crystal deposits were rare and located on the underside of the bracket (addendums Figure 4c,d). Deposits also occurred on flower-like structures (upper addendum on Figure 4c).

Results for the SS bands showed a drastic increase in the number of white precipitates (Figure 4e). The protective layer (Figure 4f) appeared similar to that observed after 7-day immersion (Figure 3f).

The surface of the ligatures changed: irregular patches were observed, indicating the formation of the protective layer (Figure 4g,h). Large amounts of debris and precipitates were present, and located on the top of the protective layer, not on the unaffected areas.

### 3.2. Semi-Quantitative Chemical Analyses of the Surface and Surface Entities

Qualitative SEM analysis of the samples revealed different types of surface deposits, precipitates, and other morphological features, in addition to an underlying fact that the surface composition can significantly differ from the nominal composition [1]. Therefore, the chemical composition of the surface entities and the unaffected surface areas were further analyzed by EDX spectroscopy.

Next to conventional presentation of the results as weight percent (wt%), the data for the relevant elements on a given location are emphasized as atomic percentages (at%), since the latter better predicts the ion release and facilitates the identification of surface deposits [1,3]. Note that the EDX is a semi-quantitative method, and it is less accurate for the lighter elements. Moreover, carbon is a special case, since it often accumulates as a contaminant on a sample surface sample [5,35,36] (or can be disregarded if the organic material is present [37]). Therefore, the wt% of C, and other lighter elements, was taken only orientationally.

#### 3.2.1. NiTi Archwires

Representative results for the surface of the NiTi archwire are shown in Figure 5. The elemental composition of the smooth unaffected regions (e.g., Spectrum 51, Figure 5a; Spectrum 52, Figure 5b; Spectrum 8, Figure 5d) showed that the ratio of Ni and Ti atomic percentages in these regions was approximately 1:1, indicating that in those areas the surface composition reflects the nominal composition of the alloy (Table 1). Moreover, comparing the relative amounts of Ni and Ti with oxygen for those locations, a very low oxygen level can be observed. Correlating this result with, for example, those reported in a study by Carroll and Kelly [5], it can be inferred that the typical oxide layer [23,24,29,38,39,40,41] did not form. This is also in accordance with the SEM results, where the protective layer would be visible due to its specific texture [24]. Note that the results showed that the oxide layer did not form even after 14-day immersion (sp. 8, Figure 5d). As indicated by SEM and confirmed by EDX, the NiTi wires used in this study were obviously depleted of the protective oxide layer, which is done by mechanical polishing [5]. Devices with a mechanically and electrolytically polished NiTi surface are also used as implants [29]. The studies showed that the prevention of Ni release by a thick oxide layer can be inadequate because it led to Ni accumulation below the protective layer, which served as a reservoir for Ni release in the case of surface damage, and led to the onset of pitting corrosion [29,42]. It has been shown that polished NiTi surfaces exhibited improved resistance to localized corrosion attacks in a chloride-rich medium [6].

On some locations, the 2:1 atomic ratio of Ti and Ni was found (sp. 50, Figure 5b). This result implied the formation of Ti_2_Ni phase, which typically occurs in NiTi wires [19,38,43,44]. Increased carbon content was observed in some locations, which could theoretically indicate the formation of the titanium carbide [38,41,45,46], but it is less likely due to the above-given reasons regarding carbon measurements with EDX. The dark grains of pure Ti, observed in some studies [6], were not found in our samples.

EDX analysis also revealed that the large crystallites observed on the NiTi surface consisted of equimolar amounts of K and Cl (sp. 51 and 53, Figure 5b; sp. 45, Figure 5c and yellow squared parts, Figure 5d), indicating the precipitation of potassium chloride crystals. In some cases (sp. 9, Figure 5d), the results implied that part of the K^+^ ions was exchanged with Na^+^. Potassium chloride is a highly soluble salt (~39 g per 100 g H_2_O at 37 °C) and was observed only on samples after immersion; hence, it can be inferred that the crystals formed after the evaporation of artificial saliva. Eliades et al. [47] reported the formation of microcrystalline KCl and NaCl deposits as well, on the surface of NiTi wires taken after 1–6 months in vivo, which substantially altered the surface composition and topography. Note that the artificial saliva used in our study contained potassium chloride and other potassium and sodium salts (vide supra, Section 2.2).

The crystals on the as-received NiTi samples contained elevated amounts of calcium ions (sp. 49, Figure 5a), accompanied by relatively high amounts of C and O, which might suggest the formation of calcium carbonate. Alongside large white precipitates, other types of deposits could be distinguished. The EDX analysis of smaller gray entities (<0.5 μm), distributed over wide areas as clusters or individually, showed equimolar amounts of Ni and Ti (sp. 52, Figure 5b) with low oxygen content, confirming them as manufacturing debris. Calcium phosphate deposits, found in previous studies after NiTi immersion in phosphate-containing solutions [6,23,48], were not detected on our samples.

Elemental analysis also showed that some of the small crystal clusters contained appreciable amounts of aluminum (e.g., Spectrum 46, Figure 5b; Spectrum 2, Figure 5c), accompanied by an increased amount of oxygen, indicating a possible formation of aluminum oxide (alumina). This result is noteworthy because aluminum was not listed in the nominal alloy composition. Yet, in several studies, some amounts of Al were found on the surface of the as-received samples [41], among the released ions in the artificial saliva and in the artificial saliva itself [49], and in the surface precipitates [25]. Moreover, Gravina et al. studied the composition of NiTi wires from six different companies and found Al_2_O_3_ in all of them, in amounts up to 5.08 wt% [50]. Other elements (contaminants) not included in the nominal alloy composition were reported for other orthodontic appliances as well [5,25].

Overall, the effect of immersion in artificial saliva on morphological changes on NiTi surface was manifested primarily through the increased number of salt deposits. This result agrees well with the study of Bobić et al. [51], where no significant changes were observed for NiTi surface roughness after 21 days of immersion.

#### 3.2.2. Stainless Steel Appliances—Brackets, Bands, and Ligatures

The SEM qualitative assessment of the surfaces of the SS brackets (Figure 1c,d, Figure 2c,d and Figure 3c,d) revealed numerous sub-micrometer-sized pits, larger irregularly shaped cavities, and large crystalline formations. EDX analysis of the unaffected areas (sp. 53, Figure 6a; sp. 12, Figure 6d) showed that the surface composition of those areas reflected the nominal alloy composition (Table 1). EDX spectra from the bottom of the large cavities (sp. 52, Figure 6a; sp. 26, Figure 6c) gave the equivalent result, which—together with visual assessment—indicated that those are not corrosion cavities, but most likely the consequence of polishing on that side of the bracket. The amounts of accompanying oxygen were such that the formation of protective oxide layer can be ruled out. After 3-day immersion, local hill-like deposits were observed (sp. 29, Figure 6b), but EDX spectra showed their composition resembles that of the unaffected surface, indicating they were not corrosion products. The expected corrosion products for steel in a solution containing hydrogen carbonate and carbonate ions, in addition to iron oxides, are carbonate species such as siderite (FeCO_3_) and magnetite (Fe_3_O_4_) [52,53], but we did not detect any of these species. Increased Al content was detected in some locations (sp. 23, Figure 6d). Large crystalline deposits, similarly to those observed on NiTi wires, contained equimolar amounts of K and Cl (Figure 6c), indicating KCl precipitation.

Some black pits contained large amounts of C (e.g., Spectrum 21, Figure 6b), indicating the formation of carbides [54]. However, in general, the EDX results of the areas encompassing black pits were not substantially different from those of the areas depleted from pits. Liu et al. [55] studied chemical composition around the pits on SS and found that the majority of pits have the identical composition to the bulk steel. Additionally, the corrosion resistance of the type of steel the brackets are made of (SS 1.4301, AISI 304) is classified as fair to good in various environments [56], but it is well known that pitting corrosion is most aggressive in solutions containing chloride ions, and frequently occurs in the oral cavity [54,57,58,59,60,61]. Hence, the observations made solely on the basis of the SEM micrographs, indicating the onset of pitting corrosion on the brackets even before the immersion in AS, cannot be rejected. Similar results were obtained for the SS bands in the pristine state (Figure 7a).

EDX analysis of the bands after immersion revealed higher amounts of oxygen compared to the brackets, indicating the formation of an oxide layer, which is consistent with the results of qualitative SEM analysis. The Al-rich deposits were also found (e.g., sp. 39, Figure 7a), but—contrarily to NiTi archwires and SS brackets—no KCl precipitates were observed. Small amounts of Ca-rich entities were found (e.g., sp. 43, Figure 7a), which might occur due to physical adsorption of calcium ions to the oxide layer [62]. An increased quantity of phosphorous atoms was observed, which could indicate the formation of phosphate deposits, which were not observed on archwires and brackets.

EDX analysis further disclosed the presence of metals not listed in the nominal composition of the bands, namely Cu (sp. 60, Figure 7d). A plausible explanation for the presence of copper might be found in the fact that Cu is a component of silver solder alloy, which is still the most commonly used soldering alloy in orthodontics [63,64]. Note that the SS bands are made of SS 1.4303 and SS 1.4551, with the latter being the filler metal. Copper(II) phosphate deposits were observed by Wendl et al. [8] on the bands with attachments from the same manufacturer.

The elemental composition of the unaffected areas on the surface of the ligatures reflected the composition of the bulk. Ca-rich deposits were again observed, but were now accompanied by increased amounts of C and O, suggesting the formation of calcium carbonate. Compared to the brackets and bands, an increased number of sites containing extraordinary amounts of carbon were observed (sp. 17 and 19, Figure 8b; sp. 6, Figure 8a; etc.), but not in the form of pits as seen on the surface of the brackets (Figure 6b). Crystals with a peculiar morphology were observed in the striation after 7-day immersion (Figure 8c). Alongside the usual bulk elements, the EDX of these crystals showed equimolar amounts of K and Cl, indicating the formation of KCl crystals also on the surface of the ligatures, but—in contrast to KCl-rich deposits on archwires and bands—their morphology on the ligatures was markedly different. A multitude of crystal morphologies has been found for salivary crystals [37], including rectangular, polyhedral, ovoid, and rod-like, with the latter describing well the morphology seen on Figure 8c.

### 3.3. Ion Release

#### 3.3.1. NiTi Archwires

The results of metal ion release from NiTi archwires immersed in artificial saliva for 3, 7 and 14 days are shown in Figure 9a. After 3 days, the relative order of eluted concentrations of the three most abundant elements in the nominal composition was preserved (Ni > Ti > Fe), but the relative amounts were not proportional; as expected, the concentration of Ni far exceeded its nominal weight percentage compared to the other ions, similar to findings of Papaionau [20]. Notably, the concentrations of Al, Cr, Zn, Sr, and Rb all exceeded the Fe concentration, even though they were not listed in the nominal composition. EDX analysis indicated the presence of aluminum oxide on the surface of the archwire (Section 3.2.1), which explains the relatively high eluted Al concentration. However, the other elements were not detected at any of the analyzed sites. The non-negligible eluted concentrations of Mn, Cu, Mo, Ba, and Pb ions were also measured (Appendix A).

From day 3 to day 7, the concentrations of all the elements increased, except those of Ti and Cr. By day 7, the Ti and Cr concentrations decreased by one-third and by an order of magnitude, respectively. With further immersion, a seemingly erratic trend in concentration change could be observed, and the trends differed for each element. This kind of erratic behavior has been reported in other studies [61]. The concentrations of some elements reached a maximum on day 7, most notably Ni. A pronounced maximum observed for Ni after 7-day immersion is consistent with previous studies [65,66], although divergent results have also been reported [8]. While Ni concentration drastically decreased by day 14, Al concentration markedly increased. The increase throughout the whole measurement period was also recorded for Fe, Zn, Pb, Sr, and Ba. Note that only Ti concentration steadily decreased.

#### 3.3.2. Stainless Steel Appliances—Brackets, Bands, and Ligatures

For all SS appliances (Figure 9b–d), Fe was the most abundant of the eluted ions, as expected. After 3 days of immersion, the relative order of eluted concentrations of the four most abundant elements in the nominal composition (Fe > Cr > Ni > Mn) was not maintained, nor were their relative amounts. Namely, Ni was detected in higher concentrations than Cr, consistent with the literature [20,67]. While an increase in eluted concentrations of all the elements apart from Ti and Cr were observed for NiTi wires from day 3 to day 7, a corresponding trend was observed only for the bands among the SS parts of the appliance. For most elements eluted from the brackets and ligatures, the values stagnated or slightly decreased during the same period. For ligatures, even the Fe concentration was lower after 7-day immersion. The trends of Ni were different for the individual parts of the appliance: a significant increase throughout the measurement period for the bands, stagnation for the ligatures, and a slight decrease on day 14 for the brackets.

Similarly to NiTi archwires, appreciable concentrations of Co, Cu, Zn, Al, Rb, Sr, and Ba were detected. Some peculiarities were also observed: the amounts of Mo found in the bands, which do not list Mo in the nominal composition, were higher than those in the brackets whose nominal composition contains 2.0–2.5 wt% Mo. An anomalously high concentration of Pb was found in the band eluate after 7-day immersion (Appendix A).

Nevertheless, by far the most prominent result was the order-of-magnitude higher concentrations of Fe and Ni ions eluted from the bands compared to all other appliances (note the values on the concentration axis in Figure 9c). Other ions followed: ≈50-fold increase in Cr concentration; ≈5-fold increase in Mn ion concentration; ≈10-fold increase in Co ion concentration; ≈10- to 100-fold increase in Cu ion concentration. Only Al concentrations were similar for bands and brackets, and lower compared to NiTi archwires. Dramatic differences in the concentrations of ions eluted from the bands compared to all other parts can be seen in Figure 10b. This figure shows the sum of concentrations of all detected ions and facilitates the comparison between used parts of fixed appliance. A 5- to 6-fold increase in aggregate concentrations was observed after 3-day immersion; an order-of-magnitude increase after 7 days; and a 4- to 5-fold increase after 14 days. Moreover, this figure shows that the bands differed from the other parts on one additional point: while the total amount of eluted ions increased steadily for archwires, brackets, and ligatures throughout the measuring period, for the bands it reached the maximum after 7-day immersion.

Figure 9 showed the eluted concentrations of the most relevant ions, not the most abundant ones. Data for all the eluted ions for each part of appliance are given in Appendix A. High concentrations of certain ions can be observed, namely, calcium, rubidium, and zinc. These ions originate from the artificial saliva itself, as shown in Figure 10a, and were therefore excluded from data shown in Figure 9.

The ion release results indicated differences between the appliances, even between those made of similar alloys, and demonstrated that the eluted concentrations of the most abundant ion for a particulate appliance (i.e., Ni for NiTi wires, Fe for SS appliances) obtained extreme values on day 7. In order to obtain additional information that could help elucidate this feature, surface roughness measurements were performed on the samples after 7 days of immersion.

#### 3.3.3. Surface Roughness

Numerous studies have shown that the corrosion resistance, and consequently the ion release from orthodontic appliances, do not correlate with surface roughness [7,9,68,69]. Due to this notion, a detailed surface roughness analysis was not the focus of this study. However, based on the results of the ion release measurements, we considered it would be informative to analyze the surface roughness of the samples after 7-day immersion. The results are shown in Figure 10. The surface roughness is reported as the arithmetic average height parameter, S_a_, a commonly used parameter which, in general, provides a good description of height variations [70]. However, for examining the corrosion effects on larger areas (note that the results are reported for a 256 × 256 μm area), the S_a_ parameter was found to be insufficient [51]; hence, the S_dr_ parameter was also used. The S_dr_ parameter denotes the developed interfacial area ratio, i.e., the percentage of additional surface area contributed by the texture as compared to an ideal plane the size of the measurement region. This parameter is affected by both the texture amplitude and the texture spacing [71].

The archwires and brackets exhibited a similar S_a_ parameter, with values that were not significantly different (Figure 11e). The values are in excellent agreement with other studies [6]. However, the S_dr_ parameter was lower for NiTi wires relative to brackets, even though not significantly. This can be attributed to the striations on the NiTi surface, which present an example of a widely spaced texture—a feature known as a characteristic that lowers the S_dr_ value.

The S_dr_ parameter revealed that the surface roughness is significantly higher for the ligatures compared to all other appliances. The ordering based on surface roughness showed no correlation with the eluted ion concentrations (Figure 10b). The ligatures, with the highest surface roughness, released on average the lowest amounts of ions. The bands, which released excessively high amounts of ions, had roughness similar to that of wires and brackets, and significantly lower roughness than ligatures. Such results imply that, in accordance with the majority of literature data, for our samples the surface roughness was not correlated to the amount of released ions.

### 3.4. Correlation of Surface Morphology Development and Ion Release

Qualitative SEM and semi-quantitative EDX analyses of NiTi archwires indicated that the immersion in artificial saliva did not result in pronounced morphological changes. Highly polished NiTi wires did not appear to contain an oxide layer in the as-received state, and the analyses did not indicate that the protective oxide layer developed during the period of immersion. However, it is known that, when formed, the protective layer is composed mostly of TiO_2_, which substantially reduces the lability of Ti [1,72]. Since the only progressive decline in eluted concentrations was observed for Ti (Figure 9a), this might suggest that a thin protective layer did start to form, but in amounts undetectable by SEM and EDX. Signs of corrosion were also not noticed, and yet the ICP-MS results showed that the eluted concentrations of all the elements varied during immersion. The maximum of Ni release after 7-day immersion is consistent with numerous studies [61,65,73,74]. The greatest difference in the amounts of released Ni and Cr, observed on day 7, is in excellent agreement with results of Barret et al. [65]. The authors provided two possible explanations for the decrease in Ni release after the initial increase: (a) the initial elution leads to depletion of Ni on the surface, which in turn leads to a decrease in the amounts of Ni released after day 7; and (b) the adherent material formed on the surface, and slowed down the Ni release. Our EDX results revealed roughly equimolar Ni:Ti composition of the NiTi surface throughout the whole immersion period; hence, the first explanation does not seem plausible for samples studied here. The Ti-oxide layer was not observed on SEM micrographs, but was implied by ICP-MS, so the second explanation seems more plausible. However, other techniques should be employed to reach more conclusive explanations.

The amounts of released chromium were at the lowest level after 14-day immersion for the SS appliances, but not for the NiTi wires. The most pronounced decrease in Cr levels was observed for the SS bands. These results implied that the part of the released Cr was used for the formation of the protective chromium-rich oxide layer [61], which was conspicuous on the SEM micrographs of SS bands, and observed on SS ligatures. It is reasonable to assume that the Fe release profiles—namely, the drop in eluted amounts after day 7 observed for brackets and bands (Figure 9b,c)—were influenced by the formation of the protective layer. However, the equivalent drop was not observed for SS ligatures, implying a different chain of chemical events.

Even though the Al ions were found in the artificial saliva before the immersion of the parts, the values of eluted concentrations were elevated by the immersion, most remarkably for NiTi archwires. The presence of Al was also detected by EDX analysis. As mentioned in Section 3.2.1, the Al ions seem to be ubiquitous in orthodontic appliances, even though Al is not listed in the nominal composition of the alloy.

An almost order-of-magnitude higher concentration of eluted ions determined for SS bands compared to all other parts could not be readily predicted from SEM and EDX measurements. Taking into account other features specific for the bands, regarding their two-alloy composition and specific manufacturing, the most plausible explanation for this result might be the onset of welding-induced corrosion. Hwang et al. [74] found that welding brought about intergranular corrosion in orthodontic bands and other parts of appliances. Park and Shearer [61] stated that the presence of additional alloys, such as silver solder, may lead to galvanic corrosion. Barret et al. [65] reported visible corrosion at localized areas adjacent to welds. Moreover, Wendl et al. [8] found a correlation between welding and the eluted amounts of Co, Cr, Mn, and Ni. Our EDX results indeed showed the increased amount of eluted Ni, Cr, and Co (in absolute values and relative to the Fe amounts; Figure 9c and Appendix A), which further corroborates the given explanation.

Although in this study the authors tried to come as close as possible to the conditions in the oral cavity, taking into account various factors, we also encountered some limitations and found some deficiencies. Under the conditions studied, all surfaces (internal and external) of each tested part of the fixed appliance are exposed to artificial saliva and are subject to the release of metal ions. This is not the case when the bracket is in the mouth, as one side is attached to the tooth and the other is exposed. The same applies to the bands. On the other hand, we could not achieve the same friction and tension under simulated conditions as in the mouth during chewing. More ions would probably be released during the intense processes in the mouth. The same is true for pH—we studied conditions simulating the patients with poor oral hygiene and low oral pH, the conditions that favor higher and more intense elution of metal ions. In addition, we must be aware that under certain circumstances there may be an accumulation of ions in the human body (due to the presence of metal ions in other sources) and significant synergistic effects that may even have an intense impact on human health, which should not be ignored and which we could not take into account in our study.

## 4. Conclusions

The main conclusions of this study can be summarized as follows:In the as-received state, four part types of a fixed orthodontic appliance exhibited dissimilar surface morphologies, which reflected variations in manufacturing processes. The onset of pitting corrosion was observed for the SS brackets and SS bands. Elemental composition of the unaffected surface areas of all parts reflected the nominal alloy composition, indicating the absence of the protective oxide layers.SEM micrographs and EDX analysis indicated that the immersion in artificial saliva did not cause the onset of pitting corrosion on NiTi archwires and SS ligatures, nor notably enhanced the corrosion process of SS brackets and SS bands during 14-day immersion. In contrast, it did result in the development of adherent layer on the surface of SS bands and SS ligatures.The immersion in artificial saliva caused the formation of crystal precipitates, mainly KCl. The morphology of KCl crystals varied for different orthodontic parts. Crystals containing Al were also detected, as well as other elements not listed in the nominal alloy composition. On the other hand, the expected corrosion products for steel immersed in a solution containing phosphate and carbonate ions—such as potassium phosphate or iron(II) carbonate (siderite)—were not detected on SS parts. The amount and average size of precipitates increased with immersion time.Ion release (ICP-MS) measurements proved to be more sensitive than SEM and EDX analyses. While latter techniques demonstrated moderate changes during immersion time, the eluted concentrations varied substantially. The release profiles for NiTi wires implied that the thin titanium oxide layer did start to form, but in amounts undetectable by SEM and EDX.Ion release was an order-of-magnitude higher for SS bands compared to the other parts of the appliance. This can most likely be assigned to the manufacturing procedure, namely welding. The welding caused increased amounts of Ni, Cr, and Co relative to eluted Fe concentration.Ion release did not correlate with surface roughness. The highest surface roughness was measured for SS ligatures, for which the lowest average amounts of eluted ions were measured. The SS bands, which released excessively high amounts of ions, had similar roughness to that of the archwires and brackets.

## Figures and Tables

**Figure 1 materials-16-04156-f001:**
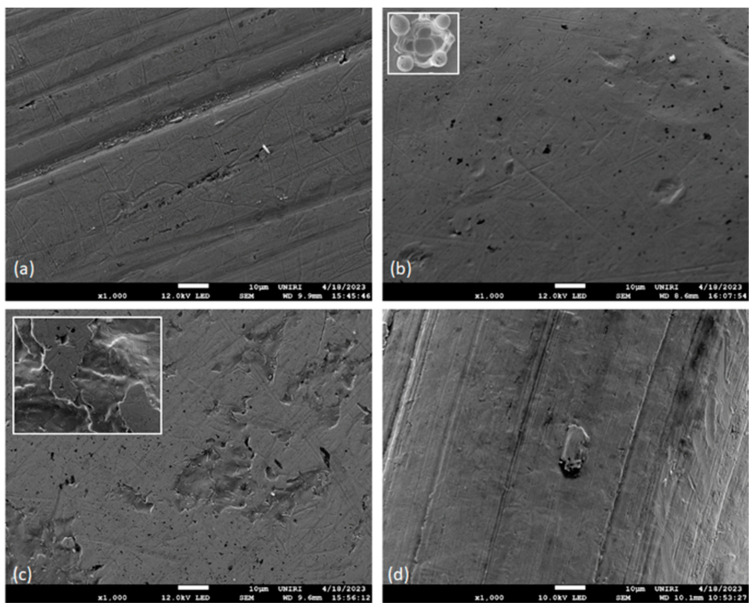
Representative SEM micrographs of (**a**) NiTi archwires, (**b**) stainless steel (SS) brackets, (**c**) SS bands and (**d**) SS ligatures, before immersion in artificial saliva.

**Figure 2 materials-16-04156-f002:**
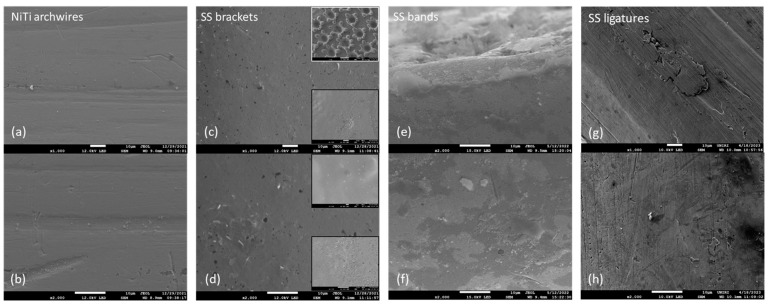
Representative lower magnification (top row) and higher magnification (bottom row) SEM micrographs of (**a**,**b**) NiTi archwires, (**c**,**d**) stainless steel brackets (SS), (**e**,**f**) SS bands, and (**g**,**h**) SS ligatures, after 3 days of immersion in artificial saliva.

**Figure 3 materials-16-04156-f003:**
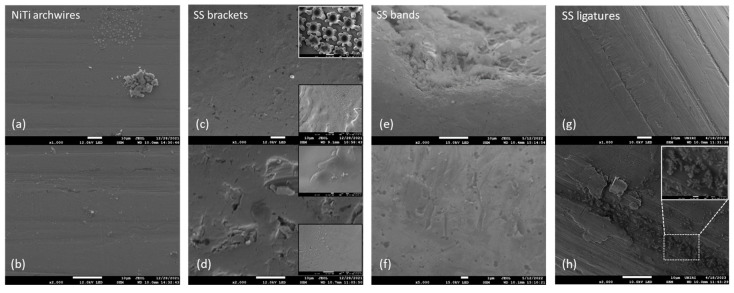
Representative lower magnification (top row) and higher magnification (bottom row) SEM micrographs of (**a**,**b**) NiTi archwires, (**c**,**d**) stainless steel brackets (SS), (**e**,**f**) SS bands, and (**g**,**h**) SS ligatures, after 7-day immersion in artificial saliva.

**Figure 4 materials-16-04156-f004:**
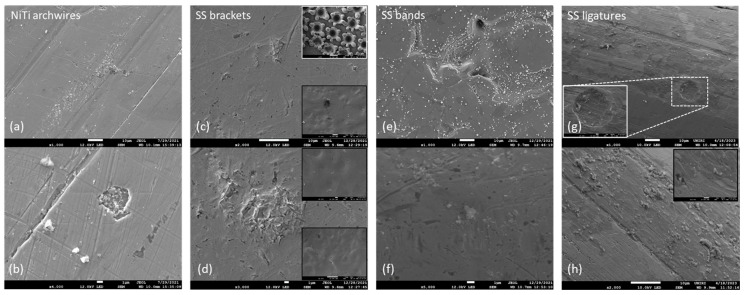
Representative lower magnification (top row) and higher magnification (bottom row) SEM micrographs of (**a**,**b**) NiTi archwires, (**c**,**d**) stainless steel (SS) brackets, (**e**,**f**) SS bands, and (**g**,**h**) SS ligatures, after 14-day immersion in artificial saliva.

**Figure 5 materials-16-04156-f005:**
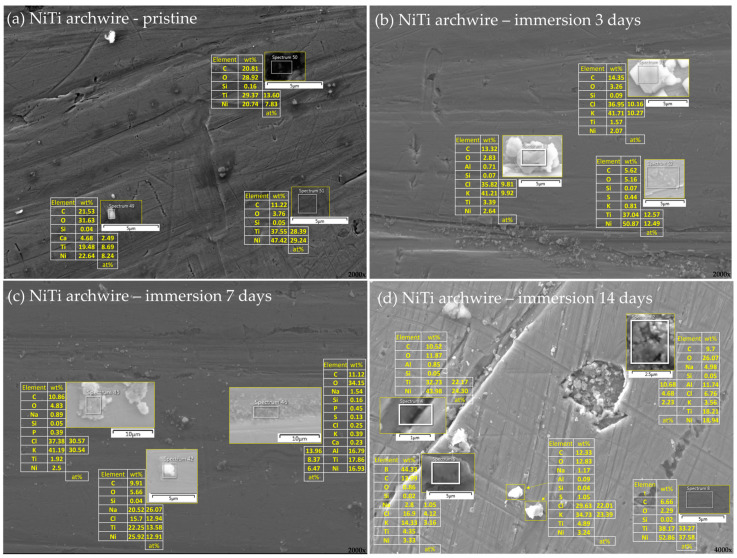
Elemental analysis of the archwire surface in an (**a**) as-received state and after (**b**) 3 days, (**c**) 7 days, and (**d**) 14 days of immersion in artificial saliva. Smaller images show a magnified view of representative surface areas/deposits/precipitates seen in the background image. The tables next to the smaller images show the respective elemental compositions, expressed as wt% for all the elements and at% for those most relevant. In the right corner of every part of a figure is a magnification.

**Figure 6 materials-16-04156-f006:**
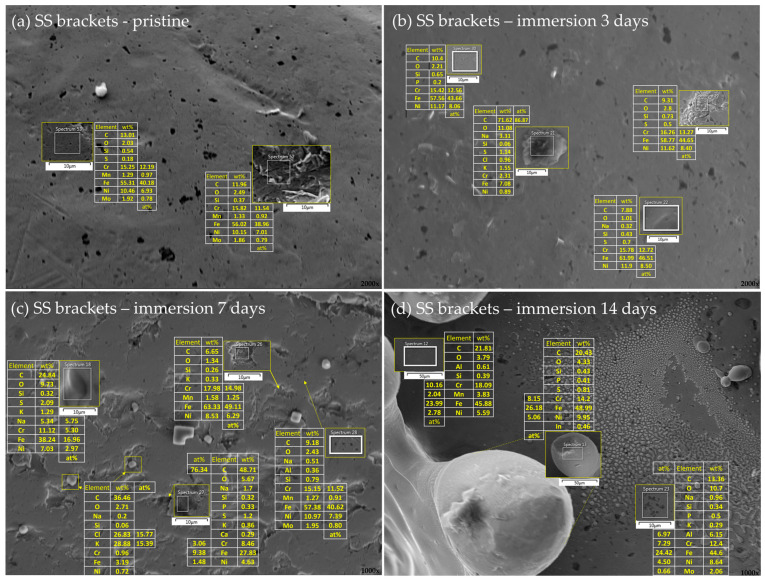
Elemental analysis of the bracket’s surface in an (**a**) as-received state and after (**b**) 3 days, (**c**) 7 days, and (**d**) 14 days of immersion in artificial saliva. Smaller images show a magnified view of representative surface areas/deposits/precipitates seen in the background image. The tables next to the smaller images show the respective elemental compositions, expressed as wt% for all the elements and at% for those most relevant. In the right corner of every part of a figure is a magnification.

**Figure 7 materials-16-04156-f007:**
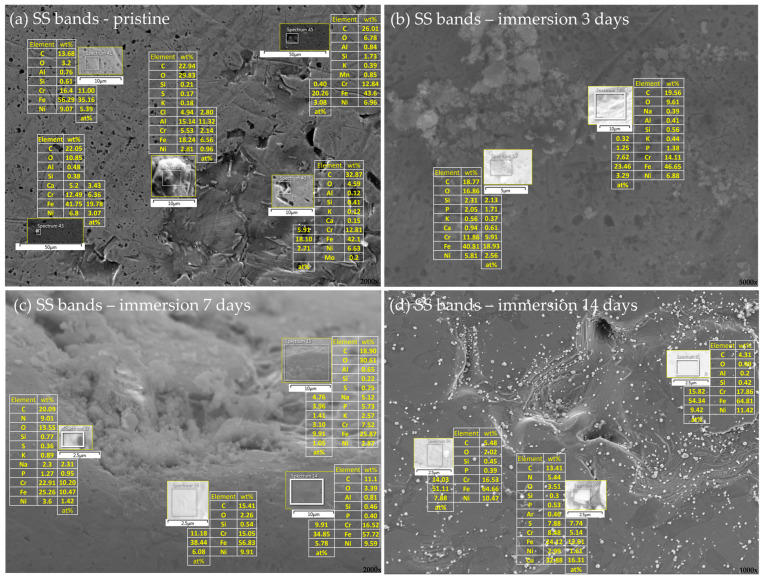
Elemental analysis of the band’s surface in an (**a**) as-received state and after (**b**) 3 days, (**c**) 7 days, and (**d**) 14 days of immersion in artificial saliva. Smaller images show a magnified view of representative surface areas/deposits/precipitates seen in the background image. The tables next to the smaller images show the respective elemental compositions, expressed as wt% for all the elements and at% for those most relevant. In the right corner of every part of a figure is a magnification.

**Figure 8 materials-16-04156-f008:**
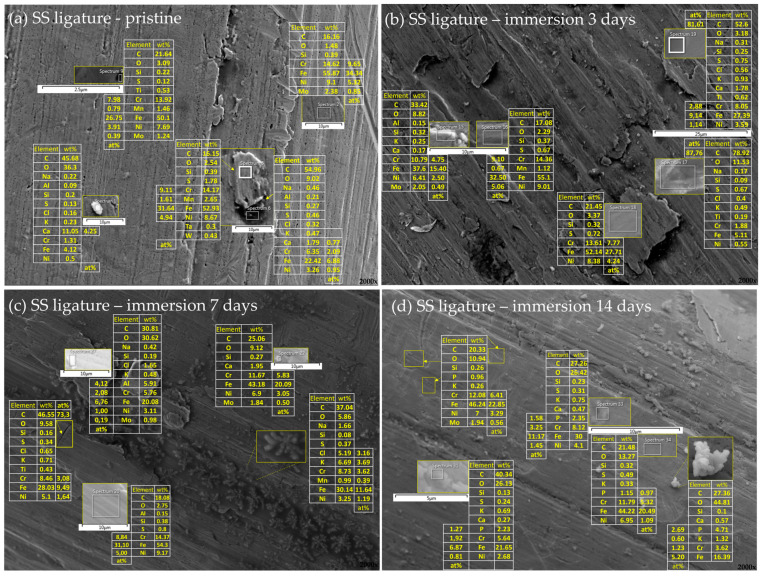
Elemental analysis of the ligatures surface in an (**a**) as-received state and after (**b**) 3 days, (**c**) 7 days, and (**d**) 14 days of immersion in artificial saliva. Smaller images show a magnified view of representative surface areas/deposits/precipitates seen in the background image. The tables next to the smaller images show the respective elemental compositions, expressed as wt% for all the elements and at% for those most relevant. In the right corner of every part of a figure is a magnification.

**Figure 9 materials-16-04156-f009:**
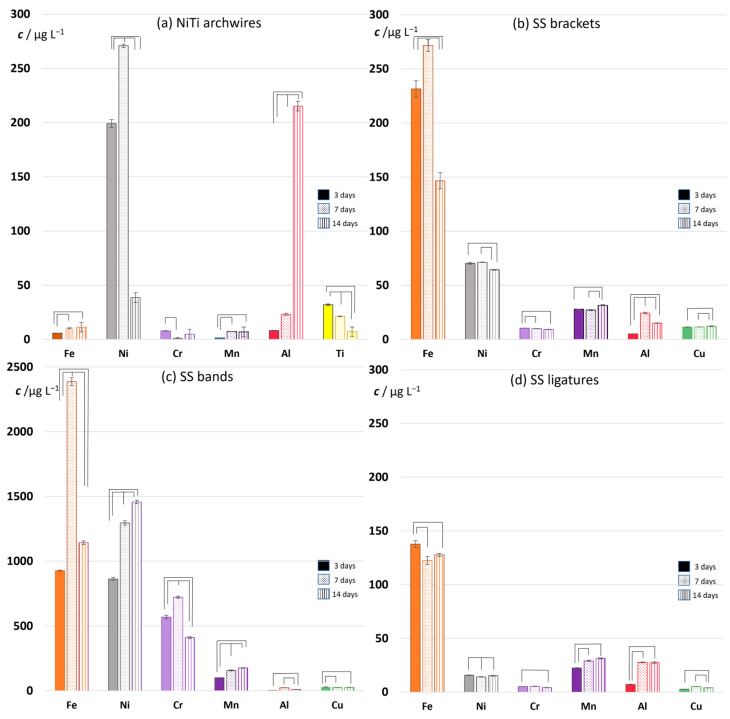
The concentrations of the most relevant ions released from (**a**) NiTi archwires, (**b**) SS brackets, (**c**) SS bands, and (**d**) SS ligatures after 3-, 7-, and 14-day immersion in artificial saliva at 37 °C. The lines/connections associate the metal ion concentrations that differ significantly (ANOVA, Post hoc Tukey HSD test, *p* < 0.05).

**Figure 10 materials-16-04156-f010:**
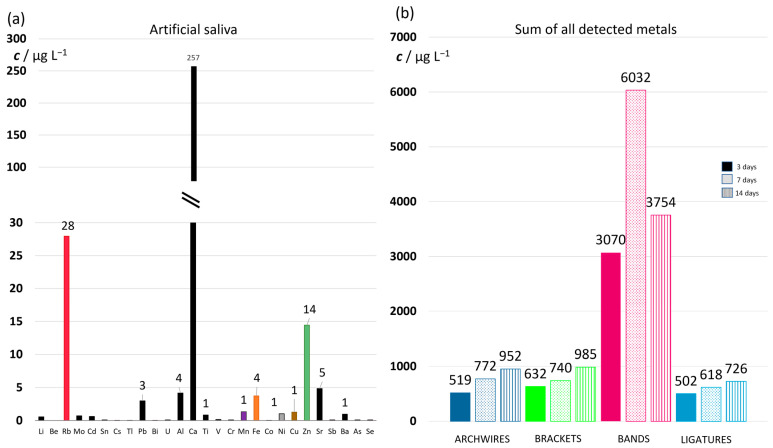
(**a**) The concentrations of the most abundant ions found in artificial saliva; (**b**) the sum of concentrations of all detected metals in archwires, brackets, bands, and ligatures after 3-, 7-, and 14-day immersion in artificial saliva at 37 °C.

**Figure 11 materials-16-04156-f011:**
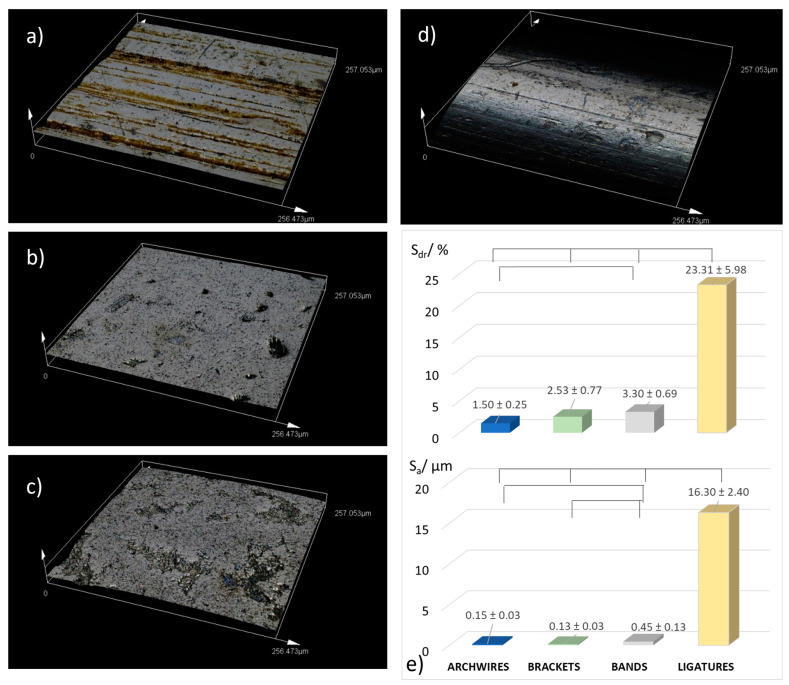
Representative CLMX images of (**a**) NiTi archwires, (**b**) SS brackets, (**c**) SS bands, and (**d**) SS ligatures, after 7-day immersion in artificial saliva. Surface roughness (**e**) is given as arithmetic mean surface roughness, S_a_, and the developed interfacial area ratio, S_dr_. The lines/connections associate the surface roughness values that differ significantly (Mann–Whitney U test, *p* < 0.05).

**Table 1 materials-16-04156-t001:** Chemical composition of parts of the fixed orthodontic appliance used in the study (NiTi archwires and SS brackets, ligatures, and bands).

	Element (wt%)	Ni	Fe	Ti	Cr	Mn	Mo	Rest
Type of appliance	Archwires	50–60	≤0.5	40–50	-	-	-	≤0.1 O, ≤0.1 Al, ≤0.1 C, ≤0.01 H, ≤0.01 N
	Brackets Ligatures	10–13	63–69	-	16.5–18.5	≤2.0	2.0–2.5	≤1 Si, ≤0.11 N, ≤0.045 P, ≤0.03 C, ≤ 0.03 S
	Bands	11–13	65–69	-	17–19	≤2.0	-	≤1 Si, ≤0.11 N, ≤0.06 C, ≤0.045 P, ≤0.03 S

## Data Availability

The data presented in this study are available on request from the corresponding author.

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
