# Peer review of "Cytotoxicity of Metal Ions Released from NiTi and Stainless Steel Orthodontic Appliances, Part 1: Surface Morphology and Ion Release Variations"

_materials, 2023, doi:10.3390/ma16114156_

Round 1

Reviewer 1 Report

This is a very interesting article.Congratulations to the authors.

I suggest the authors in figures 5-8 to include bigger images because the values are hard to distinguish.

Could the authors please specify the limitations of the present study?

Just moderate revision.

Author Response

The authors are very grateful for your review and your useful comments and suggestions.

I suggest the authors in figures 5-8 to include bigger images because the values are hard to distinguish.

In all figures 5-8 font is corrected to 14.

Could the authors please specify the limitations of the present study?

Limitations of the study are added at the end of discussion part.

Following your suggestions and the other reviewers, Abstract, and Introduction are modified/improved, and English language edited throughout the manuscript.

Reviewer 2 Report

The authors presented research on the experimental investigation of ion release of orthodontic elements in artificial saliva using ICP-MS, EDX/SEM, and surface roughness measurement techniques. The research provides a useful contribution to the knowledge in this field. Some minor comments which authors may address for bringing clarity to the manuscript.

1. What is the minimum resolution of the confocal microscope used for surface roughness measurements? There seems to be not much variation in the roughness parameters before and after the immersion tests. Does that mean, the instrument couldn't capture the variation of these parameters despite ion release? Does AFM help to capture these variations? Please comment.

2. What is the reason for EDX analysis when there is not much variation in the surface roughness? It is apparent that the SEM is not providing any value here! Is XPS not helpful to obtain more information on chemical changes that occur on the surfaces? Please comment.

3. The data representation of EDX analysis is not so clear. Could authors assimilate the data presented in Figs 5-8 in tabular form? 

n-
20ductively coupled plasma mass spectrometry

n-
20ductively coupled plasma mass spectrometry

The manuscript is well-written and the language is adequate for publishing in this journal. 

Author Response

The authors are very grateful for your review and your useful comments and suggestions.

  1. What is the minimum resolution of the confocal microscope used for surface roughness measurements? There seems to be not much variation in the roughness parameters before and after the immersion tests. Does that mean, the instrument couldn't capture the variation of these parameters despite ion release? Does AFM help to capture these variations? Please comment.

The maximum lateral resolution of the confocal laser scanning microscope (CLSM) used was 0.12 μm.

Yes, you are correct, atomic force microscopy would probably show greater variations. We decided to use CLSM from the reason of the equipment availability and the possibility to scan and analyse larger areas (2, 640 µm2), as in case of e.g. AFM (only several nm2).

  1. What is the reason for EDX analysis when there is not much variation in the surface roughness? It is apparent that the SEM is not providing any value here! Is XPS not helpful to obtain more information on chemical changes that occur on the surfaces? Please comment.

Indeed, XPS can be very useful to obtain information about chemical bonds, but this information refers to a larger part of the sample (about 2 mm) and this was not our intention.

Since SEM is equipped with an EDS system, it allows elemental analysis of features seen on the SEM monitor. We consider this ability to obtain elemental composition in a defined area (from a specific detail on the surface) to be a major advantage for this study, and in general EDS is used extensively for the study of metallic and ceramic samples and their associated weathering crusts or corrosion products.

The detection limit in EDS depends on the surface condition of the sample, the smoother the surface, the lower the detection limit.

  1. The data representation of EDX analysis is not so clear. Could authors assimilate the data presented in Figs 5-8 in tabular form? 

There is a huge amount of SEM, and EDX data. We were looking for an appealing form of presentation to summarise as much relevant data as possible in one figure so that it could be read and understood fluently. The authors found the chosen form of presentation very informative and concise and would like to keep it that way. However, we have taken into account that the images are not clear, so in all Figures 5-8 the font size has been corrected to 14. And the figures can be larger.

If the changes made are not enough, we can prepare additional tables instead of presenting tabular data on the figures.

Following suggestions of the other reviewers, Abstract, and Introduction are modified/improved, and English language edited throughout the manuscript.

Reviewer 3 Report

The authors have conducted a detailed comparative study of ion release from four common orthodontic devices, characterizing them mainly in terms of microscopic features, elemental analysis, and surface roughness, with more systematic and detailed experiments. I suggest that this manuscript could be accepted after major revision. Some specific comments are listed as follows:

1For orthodontics (also mentioned in the title), I think cytotoxicity experiments are necessary to demonstrate their feasibility for biomedical use. Perhaps the authors did not cover this in the first part.

2From a biomedical engineering point of view, it is recommended that a study be conducted to determine whether this oral medical device complies with national (European, American, Chinese, etc.) standards.

3The experiment focused on the effect of different immersion cycles. For the oral cavity, does the pH also have an effect on ion release? Should it be considered as one of the control factors? (As in Figure 10, only the human body temperature of 37°C was considered).

Minor editing of English language required

Author Response

The authors are very grateful for your review and your useful comments and suggestions.

The authors have conducted a detailed comparative study of ion release from four common orthodontic devices, characterizing them mainly in terms of microscopic features, elemental analysis, and surface roughness, with more systematic and detailed experiments. I suggest that this manuscript could be accepted after major revision. Some specific comments are listed as follows:

1. For orthodontics (also mentioned in the title), I think cytotoxicity experiments are necessary to demonstrate their feasibility for biomedical use. Perhaps the authors did not cover this in the first part.

In our project, we address the cytotoxicity of metal ions released from fixed orthodontic appliances made of NiTi archwires and stainless steel brackets, bands and ligatures. In the submitted manuscript, we address the surface morphology and variations in ion release from these orthodontic appliance components. This is the first part of the research as we investigate the metal content and composition in eluates prepared from orthodontic appliance components in artificial saliva. These eluates will be used for toxicity testing on different types of cell lines and on yeast. We are currently revising a paper presenting the results of toxicity tests on cells of the gastrointestinal tract (Part 2 of the project results).

If the reviewer disagrees, we can change the title of this paper to: Qualitative and quantitative surface analysis of orthodontic appliances (or sthg similar).

2、From a biomedical engineering point of view, it is recommended that a study be conducted to determine whether this oral medical device complies with national (European, American, Chinese, etc.) standards.

Yes, you are absolutely right. We prepared the samples (eluates in artificial saliva) according to current International Organization for Standardization (ISO) Standard (ISO 10993-5:2009). This standard describes test methods to assess the in vitro cytotoxicity of medical devices.

We accidentally omitted this, so it is inserted in the manuscript.

3、The experiment focused on the effect of different immersion cycles. For the oral cavity, does the pH also have an effect on ion release? Should it be considered as one of the control factors? (As in Figure 10, only the human body temperature of 37°C was considered).

Yes, you are absolutely right. Based your comment/request we added this part in the Introduction:

Under normal circumstances, the pH in the oral cavity is not stable and changes according to the intake of food and drink, but homeostasis returns to neutral values. Maintaining oral hygiene is extremely difficult during fixed orthodontic therapy. The accumulation of plaque and inflammatory processes in the gingiva lead to a drop in pH and the formation of an acidic environment that promotes the degradation of dental materials [doi:10.3390/ijerph18116049]. For this reason, it is interesting to study the oral conditions in patients with poor hygiene.

….and we decided to do precisely that.

Following your suggestions and of the other reviewers, Abstract, and Introduction are modified/improved, deficiencies of the study added, presentation of the results clarified and English language edited throughout the manuscript.

Reviewer 4 Report

It has a very accessible introduction for the dentist, and it is clear, but it lacks showing the importance of personal predisposition and individual vulnerability.

It would be important to highlight, in the introduction, that this topic is of great importance, since it is in the child population, most vulnerable to toxic effects, those who are most exposed to being treated mainly by orthodontics in general at an early age.

In the conclusions, the clinic effects should also be highlighted more and especially that of personal vulnerability.

The work is carried out with rigor, as they say there have been studies of this type for years, but they have used current methodologies and I think it supports existing knowledge, however, it should place more emphasis on personal vulnerability.

Author Response

The authors are very grateful for your review and your useful comments and suggestions.

It has a very accessible introduction for the dentist, and it is clear, but it lacks showing the importance of personal predisposition and individual vulnerability.

We have added a whole new section to the introduction explaining the other factors that contribute to the elution of metal ions into the oral cavity during orthodontic therapy. Then we listed the side effects and explained the contribution of pH to the degradation of dental materials.

To say more about this is beyond the scope of this article.

In another article, we investigated the toxicity of eluted metal ions in artificial saliva and the response of cells of the gastrointestinal tract to metal ions and explained the toxicity of each eluted metal individually.

It would be important to highlight, in the introduction, that this topic is of great importance, since it is in the child population, most vulnerable to toxic effects, those who are most exposed to being treated mainly by orthodontics in general at an early age.

We agree with you that this issue is of great importance because fixed orthodontic appliances are mainly used by the most vulnerable population, our children.

In the conclusions, the clinic effects should also be highlighted more and especially that of personal vulnerability.

Since we are not addressing personal risk here, it is not appropriate to draw a conclusion in this part. However, considering your comment in the part where we discussed the shortcomings of this study, we consider the "personal touch in the whole story" (patients with poor oral hygiene and the synergistic effect of metals accumulated in the organism from different sources). We hope that with that we will meet your expectations.

The work is carried out with rigor, as they say there have been studies of this type for years, but they have used current methodologies and I think it supports existing knowledge, however, it should place more emphasis on personal vulnerability.

Thank you for your comment - indeed, we have used the current methodology to study things that are already well researched, but we are trying to shed a whole new light on them.

Following your suggestions and of the other reviewers, abstract, and an Introduction are modified/improved, deficiencies of the study added, presentation of the results clarified, conclusions modified and English language edited throughout the manuscript.

Reviewer 5 Report

Dear authors, thank you for your submission. The manuscript has a significant scientific relevance; however, I indicate some corrections to be made before publication.

The abstract lacks an organized structure with a lack of coherence. Include the objectives of the study. Consider improving the conclusion.

Ion elution, surface morphology, NiTi archwires are not MeshTerms. Consider adapting them.

Table 1: consider adding caption.

How were the EDX analysis readings standardized for each sample? From the representations in figures 5-8, the choices seem to be random or chosen, which may characterize a methodological bias.

Surface roughness: even though this analysis is not considered the objective of the research, but it was carried out and, therefore, must be analyzed appropriately. Consider applying statistical analysis and discuss it accordingly. This adequacy is necessary as it is part of the conclusion of your study.

Author Response

The authors are very grateful for your review and your useful comments and suggestions.

The abstract lacks an organized structure with a lack of coherence. Include the objectives of the study. Consider improving the conclusion.

Per your remark, the abstract is now amended to better reflect the structure suggested in the journal template [(1) Background: Place the question addressed in a broad context and highlight the purpose of the study; (2) Methods: briefly describe the main methods or treatments applied; (3) Results: summarize the article’s main findings; (4) Conclusions: indicate the main conclusions or interpretations.] The objectives are now emphasized. The Conclusion section is also revised and improved, and we thank you for your suggestion.

Ion elution, surface morphology, NiTi archwires are not MeshTerms. Consider adapting them.

Following your suggestion, we adapted the keywords to improve the match with MeshTerms whenever possible (namely: „surface morphology“ to „surface properties“, „NiTi archwires“ to just „NiTi“, because „copper NiTi“ – available at MeshTerms – is not applicable here; „elution“ was available as „drug-elution“, which was also not applicable here, hence we could not adapt this keyword).

Table 1: consider adding caption.

Thank you for the comment. Caption is improved.

How were the EDX analysis readings standardized for each sample? From the representations in figures 5-8, the choices seem to be random or chosen, which may characterize a methodological bias.

Thank you for this comment.

For each sample electron image, spectrum and composition of determined metals was obtained (please see the attach).

From huge amount of data, we try to present only the most important and the most relevant.

Per your remark, a sentence explaining standardization is added to the manuscript (Methods part). Every element is standardized using standard lebel (for aluminum Al2O3 is used, for argon Ar(V), for calcium Wollastonite, for carbon vitamin C, for chlorine NaCl, for chromium Cr, for copper Cu, for iron Fe, for magnesium MgO, for manganum Mn, for nickel Ni, for phosphorous GaP, for oxygen and for silicon SiO2, for potassium KBr, for radon, Rh, and for sodium Albite)

Surface roughness: even though this analysis is not considered the objective of the research, but it was carried out and, therefore, must be analyzed appropriately. Consider applying statistical analysis and discuss it accordingly. This adequacy is necessary as it is part of the conclusion of your study.

The statistical analysis is now done, with the Figures and captions changed accordingly, and the subsequent text accordingly modified.

Following your suggestions and of the other reviewers, Abstract, and Introduction are modified/improved, deficiencies of the study added, presentation of the results clarified, conclusions improved and English language edited throughout the manuscript.

Reviewer 6 Report

The reviewer really appreciates the efforts of the authors to conduct this study. The study design is good enough to extract valuable conclusions. The manuscript is well written without leaving any major issues. The reviewer would like to suggest some minor corrections in the manuscript.

·       Please add a reference for the composition of artificial saliva used in this experiment

·       In Figure 9 the author can add a statistical comparison between different time periods in amount of ion release from each group

·        Please increase the font size and resolution of Figure 9

·       Please increase the font size and resolution of Figure 10. The horizontal and vertical axis is difficult to read.

·       Add statistical comparison for the quantitative data shown in Figure 11

·       Please revise the references. In some references, the year of publication is not in bold

·       The author can eliminate some old references where multiple citations were used for a single statement (just a suggestion)  

Author Response

The authors are very grateful for your review and your useful comments and suggestions.

  • Please add a reference for the composition of artificial saliva used in this experiment

Thank you for the comment. Reference is added.

  • In Figure 9 the author can add a statistical comparison between different time periods in amount of ion release from each group

The statistical analysis is now done, with the Figures and captions changed accordingly.

  • Please increase the font size and resolution of Figure 9.

Thank you for the comment. Hope we improve that sufficiently.

  • Please increase the font size and resolution of Figure 10. The horizontal and vertical axis is difficult to read.

Thank you for the comment. Hope we improve that sufficiently.

  • Add statistical comparison for the quantitative data shown in Figure 11

The statistical analysis is now done, with the Figure and caption changed accordingly.

  • Please revise the references. In some references, the year of publication is not in bold

 The references are revised, and years of publications are all bolded, thank you for noticing.

  • The author can eliminate some old references where multiple citations were used for a single statement (just a suggestion)  

Per your suggestion, we revised the citations and eliminated some older references in cases of multiple citations for a single statement, whenever it was possible without losing relevant information (namely, references Lee et al 2001, Zhang et al 2006, Lin et al 2009 and Brantley et al 2008 were eliminated). In some cases, where we state that “numerous studies have shown”, multiple citations are left unaltered, in order to really corroborate our claim; we hope it is all right.

Please note that new references, regarding newly added parts of text, had to be added due to request of other reviewers.   

Following your suggestions and of the other reviewers, Abstract, and Introduction are modified/improved, deficiencies of the study added, presentation of the results clarified and English language edited throughout the manuscript.

Round 2

Reviewer 3 Report

Accept in present form